# Epithelial-to-Mesenchymal Transition Drives Invasiveness of Breast Cancer Brain Metastases

**DOI:** 10.3390/cancers14133115

**Published:** 2022-06-25

**Authors:** Andreia S. Margarido, Rebeca Uceda-Castro, Kerstin Hahn, Roebi de Bruijn, Lennart Kester, Ingrid Hofland, Jeroen Lohuis, Danielle Seinstra, Annegien Broeks, Jos Jonkers, Marike L. D. Broekman, Pieter Wesseling, Claire Vennin, Miguel Vizoso, Jacco van Rheenen

**Affiliations:** 1Division of Molecular Pathology, Oncode Institute, The Netherlands Cancer Institute, 1066 CX Amsterdam, The Netherlands; a.da.cruz.margarido@nki.nl (A.S.M.); r.uceda.castro@nki.nl (R.U.-C.); kerstin.hahn@roche.com (K.H.); ro.d.buijn@nki.nl (R.d.B.); L.A.Kester@prinsesmaximacentrum.nl (L.K.); j.lohuis@nki.nl (J.L.); j.jonkers@nki.nl (J.J.); c.vennin@nki.nl (C.V.); 2Division of Molecular Carcinogenesis, Oncode Institute, The Netherlands Cancer Institute, 1066 CX Amsterdam, The Netherlands; 3Core Facility Molecular Pathology & Biobanking, The Netherlands Cancer Institute, 1066 CX Amsterdam, The Netherlands; i.hofland@nki.nl (I.H.); a.broeks@nki.nl (A.B.); 4Department of Pathology, Amsterdam University Medical Centers/VUmc and Brain Tumor Center Amsterdam, 1081 HV Amsterdam, The Netherlands; daantjeseinstra@hotmail.com (D.S.); p.wesseling@amsterdamumc.nl (P.W.); 5Department of Neurosurgery, Leiden University Medical Center, 2333 ZA Leiden, The Netherlands; m.l.d.broekman-4@umcutrecht.nl; 6Department of Neurosurgery, Haaglanden Medical Center, Lijnbaan, 2512 VA The Hague, The Netherlands; 7Department of Neurology, Massachusetts General Hospital, Harvard Medical School, Boston, MA 02115, USA; 8Laboratory for Childhood Cancer Pathology, Princess Máxima Center for Pediatric Oncology, 3584 CS Utrecht, The Netherlands

**Keywords:** epithelial-to-mesenchymal transition, breast cancer, brain metastases, invasion, surgical resection, recurrence, intravital imaging

## Abstract

**Simple Summary:**

The prevalence of breast cancer brain metastasis has increased over the last decades, yet those patients are considered untreatable. Surgical removal of brain metastases is impeded by the presence of infiltrative tumor cells; however, the identity and behavior of those cells remain understudied. We demonstrate that cancer cells that invade the brain parenchyma are in a mesenchymal state and undergo EMT. We also demonstrate that removing those infiltrated tumor mesenchymal cells improves the efficacy of surgery and opens new avenues to better treat patients.

**Abstract:**

(1) Background: an increasing number of breast cancer patients develop lethal brain metastases (BM). The complete removal of these tumors by surgery becomes complicated when cells infiltrate into the brain parenchyma. However, little is known about the nature of these invading cells in breast cancer brain metastasis (BCBM). (2) Methods: we use intravital microscopy through a cranial window to study the behavior of invading cells in a mouse model of BCBM. (3) Results: we demonstrate that BCBM cells that escape from the metastatic mass and infiltrate into brain parenchyma undergo epithelial-to-mesenchymal transition (EMT). Moreover, cells undergoing EMT revert to an epithelial state when growing tumor masses in the brain. Lastly, through multiplex immunohistochemistry, we confirm the presence of these infiltrative cells in EMT in patient samples. (4) Conclusions: together, our data identify the critical role of EMT in the invasive behavior of BCBM, which warrants further consideration to target those cells when treating BCBM.

## 1. Introduction

Although early detection of primary tumors and novel treatment strategies have improved outcomes of patients with breast cancer (BC), an increasingly large number of BC patients live long enough to develop lethal BM: 30–50% of patients with Her2+ BC and 25–45% of triple negative BC develop brain metastasis [1,2,3,4,5]. Standard-of-care treatments for BCBM include neurosurgical resection, systemic therapies and stereotactic radiotherapy. However, these treatments largely fail to improve patients’ outcomes and BCBM remains a leading cause of BC-related death [5]. In particular, the ability of BCBM cells to infiltrate into the healthy brain tissue renders BCBM difficult to treat. Although tumor resection may seem complete by magnetic resonance imaging (MRI), the limited resolution of this technique does not allow the detection of individual cells that have invaded beyond the resection cavity. Consequently, overall survival rates following surgical removal of BCBM remain dismal, with 60% of patients with confirmed complete resection by MRI locally recurring within 1 year after surgery [6,7]. Nevertheless, the properties of BCBM cells that invade the brain parenchyma and drive patients’ relapse remain under-studied. It is therefore critical to better understand the mechanisms that drive the invasive behavior of BCBM cells in order to design efficient treatment strategies for this dreadful disease.

The migratory and infiltrative behaviors of BC cells that escape primary tumors have been extensively studied [8,9,10,11,12,13]. It is generally accepted that in order to become invasive, cells in the primary tumor hijack the developmental program known as EMT [14,15,16,17,18,19,20,21,22,23,24]. For instance, we and others previously used intravital microscopy to show that cells that escape from primary breast tumor and invade the healthy stroma undergo EMT [25,26,27,28]. During EMT, cells downregulate the expression of epithelial markers such as the cell–cell adhesion molecule E-cadherin (E-cad), thereby losing cell polarity and cell–cell adhesion. In parallel, cells undergoing EMT acquire mesenchymal markers that drive stemness and invasiveness such as Zeb1 and Twist [14,18,21,29,30,31,32,33,34]. To date, the role of EMT in BC has mainly been studied in the context of cancer cells disseminating from the primary tumor and during the outgrowth of extracranial metastases, but the existence and functions of EMT in BCBM remain unknown.

We previously used the mouse mammary tumor virus polyoma middle-T antigen (PyMT) model crossed with fluorescent E-cadherin-mCFP (E-cad-mCFP) reporter mice to report EMT in ductal mammary carcinomas and lung metastases [26,27]. Here, we employed this EMT reporter mouse to generate a BCBM model in order to image, isolate and characterize cells that have undergone EMT in the context of brain metastasis. Using intravital imaging, we identify infiltrative tumor cells in BCBM which preclude the complete removal of the lesion by surgery, leading to increased BCBM recurrence in mice. Moreover, we demonstrate that in contrast to cells in the tumor mass, cells infiltrating into the brain tissue display a mesenchymal expression profile. Additionally, we find that mesenchymal cells can revert to an epithelial state during outgrowth of tumor masses in the brain. Importantly, using multiplex immunohistochemistry, we also detect this infiltrating population of cells in EMT in the brain tissues of BCBM patients, illustrating the clinical relevance of our observations. Collectively, our data warrants further consideration to target EMT as part of the treatment of BCBM.

## 2. Materials and Methods

### 2.1. Mice

All experiments were performed under the guidelines of the Animal Welfare Committee of the Royal Netherlands Academy of Arts and the Netherlands Cancer Institute, in accordance with national guidelines. Animals were kept at the Hubrecht animal facility in Utrecht, the Netherlands or at the Netherlands Cancer Institute facility in Amsterdam, the Netherlands. All animals were housed in individually ventilated cages under specific-pathogen-free conditions and received chow and water ad libitum. Female ROSA26^mTmG^ C57BL/6J mice were purchased from Jackson Laboratory and backcrossed to FVB mice at the animal facility of the Hubrecht and of the Netherlands Cancer Institute. Female FVB mice were acquired from Janvier. All mice were injected with tumor cells when they were 8–20 weeks old.

### 2.2. Serial Intracranial Injections to Generate the BCBM Model

End-stage primary breast tumors were isolated from PyMT E-cad-mCFP mice [27] and frozen in freezing medium (Gibco, Cat. No. 12648-010, ThermoFisher Scientific, Waltham, MA, USA). For the first intracranial injection, tumors were thawed and minced on ice with sterile knives and enzymatically digested at 37 °C for 20–30 min while shaking at 900 rpm. The digestion mix was composed of 0.01 g collagenase A (Roche, Cat. No. 10103586001, Basel, Switzerland), 2 mL TrypLE Express (Gibco, Cat. No. 12605010) and 3 mL DMEM/F12 GlutaMAX (Gibco, Cat. No. 10565018) supplemented with 10 mmol/L Hepes (Gibco, Cat. No. 15630106), 100 μg/mL streptomycin and 100 U/mL penicillin (Gibco, Cat. No. 15140122). Undigested tumor pieces were removed by spinning down at 800 rcf at 4 °C and for 5 min. Then, 10 μg/mL DNase I (Roche, Cat. No. 10104159001) in DMEM/F12 GlutaMAX medium were used to dissolve the pellet while manually shaking the tube for 5 min. The final pellet was resuspended in 3 μL per intracranial injection of PBS. After a tumor was formed in the brain of the recipient mouse, the tumor was isolated and cut into smaller tumor pieces prior to freezing (Gibco, Cat. No. 12648-010). This whole process is considered as one enrichment round in the brain. We repeated this process for a total of 6 times using the tumor pieces from the previous serial enrichment round. At the 6th enrichment round, we derived organoids which are described in this study as ‘BCBM organoids’.

### 2.3. Organoids Culturing

PyMT E-cad-mCFP organoids were cultured in drops of 50 μL of Basal Membrane Extract (BME) type 2 (R&D systems, Cat. No. 3533-005-02) diluted at a 2:1 ratio with DMEM/F12 GlutaMAX medium (Gibco, Cat. No. 10565018) in a pre-warm 24-well plate (Greiner bio-one, Cat. No. 662160) and inverted while solidifying for 30 min at 37 °C, 5% CO_2_. Organoids were cultured in complete DMEM/F12 GlutaMAX medium (Gibco, Cat. No. 10565018), supplemented with 10 mmol/L Hepes (Gibco, Cat. No. 15630106), 100 μg/mL streptomycin, 100 U/mL penicillin (Gibco, Cat. No. 15140122), 10.08 ng/mL FGF (Gibco, Cat. No. PHG0261) and 1× B27 supplement (Gibco, Cat. No. 17504001) and incubated at 37 °C, 5% CO_2_. To confirm Mycoplasma-free culturing of the organoids, the MycoAlert PLUS Kit (Lonza Cat. No. LT07-118, Basel, Switzerland) was routinely used according to the manufacturer’s protocol.

### 2.4. Lentivirus Production and Transduction

Generation of the PyMT E-cad-mCFP; H2B-Dendra2 line was performed using the vector plvx-UBC-H2B-Dendra2-puromycin. Human embryonic kidney (HEK) 293T cells at a confluence of 80% were used for lentivirus production. Per 10 cm dish (Greiner, Cat. No. 664160) of HEK 293T cells, 7.5 µg of psPAX2, 2.5 µg PMD2.G and 10 µg of our H2B-Dendra2 construct were mixed in 1 mL Opti-MEM (ThermoFisher Scientific, Cat. No. 31985070, Waltham, MA, USA). After mixing, 1 mL of Opti-MEM with 40 µL lipofectamine 2000 (ThermoFisher Scientific, Cat. No. 11668019) were added to the plasmid mix and incubated at room temperature for 15 min. The mix was carefully added to the HEK 293T cells. The following morning, medium was refreshed with DMEM GlutaMAX (Gibco, Cat. No. 31966047) supplemented with 100 μg/mL streptomycin, 100 U/mL penicillin (Gibco, Cat. No. 15140122). After 48 h, medium was collected and filtered through a 0.22 µm filter (Milipore, Cat. No. SLGS033SS, Burlington, MA, USA). Filtered medium was concentrated with an Amilcon Ultra-15 10 k column (Milipore, Cat. No. UFC905024) for 1 h at 4000 rcf. Organoids were trypsinized into smaller clusters of approximately 8 cells and incubated with 250 µL virus, 100 µg/mL polybrene (Sigma, Cat. No. TR-1003-G, St. Louis, MO, USA) and 10 µmol/L Y-27632 (Bio Connect, Cat. No. S1049) on a 48-well plate low adherence (Greiner, Cat. No. 677970). Spin infection was performed at 36 ºC, 600 rcf for 1 h and organoids were subsequently incubated at 37 °C for 6 h. Organoids were washed twice with DMEM/F12 GlutaMAX medium (Gibco, Cat. No. 10565018) and plated in BME. Complete DMEM/F12 GlutaMAX medium (Gibco, Cat. No. 10565018), supplemented with 10 mmol/L Hepes (Gibco, Cat. No. 15630106), 100 μg/mL streptomycin, 100 U/mL penicillin (Gibco, Cat. No. 15140122), 10.08 ng/mL FGF (Gibco, Cat. No. PHG0261), 1xB27 supplement (Gibco, Cat. No. 17504001) and 10 µmol/L Y-27632 (Bio Connect, Cat. No. S1049) was added to the organoids for 2 days maximum. Organoids selection was performed with 0.5 μg/mL puromycin (Gibco Cat. No. A1113803).

### 2.5. shRNAs Constructs Selection, Viruses Production and Transduction

ARPC3 knockdown experiments were performed with shRNAs provided by The RNAi Consortium (TRC) mouse library. Their oligoIDs were the following: shRNA#1-TRCN0000110780, shRNA#2-TRCN0000110781, shRNA#3-TRCN0000110783 and shRNA#4-TRCN0000110784. Resistance cassette from these vectors was changed from puromycin to blasticidin and control pLKO.1-scramble-blasticidin was purchased from Addgene (plasmid 26701, Watertown, MA, USA). The puromycin cassette for the shARPC3 constructs was removed using restriction enzymes BamH1 and Kpnl. Concentrated viruses were produced as described in [35] by co-transfection of 4 plasmids: VSV-G (9 µg), pMLD (12.5 µg), kREV (6.25 µg) and shARPC3 (32 µg) in HEK 293T cells. Organoid transduction was performed as described in ‘Lentivirus production and transduction’. Virus titers were calculated with a qPCR lentivirus titration titer kit (abm, Cat. No. LV900, Richmond, BC, Canada) as per the manufacturer’s protocol. Organoids were transduced with a multiplicity of infection (MOI) of 10. To determine the volume of virus to use during transductions, we used the formula: MOI = number of viral units/number of cells. Transduced organoids were selected with 10 µg/mL of blasticidin (Gibco, Cat. No. A1113903) and kept in culture with this selection.

### 2.6. Cranial Window Implantation and BCBM Injection under the Window

BCBM derived organoids were digested with TrypLE Express (Gibco, Cat. No. 12605010) for ~20 min at 900 rpm and 37 °C to produce a single cell suspension. Digestion was stopped by adding DMEM F12 medium. Per injection, 100,000 cells were resuspended in 3 μL of PBS and injected just before the implantation of the cranial window. Cranial window implantation was performed as previously described [36]. In summary, mice were treated with 0.067 mg/mL rimadyl (carprofen, Zoetis) in drinking water 1 day before injection and for 3 days post surgery. Furthermore, 30 min before surgery mice were subcutaneously injected with 0.1 mg/kg temgesic (buprenorphine, Indivior Europe Limited) and 8–12 h after window implantation. Mice were anesthetized via inhalation of 2% (*v*/*v*) isoflurane and eyes were covered with duratears (Alcon). Once placed in a stereotactic frame, the mouse head was shaved and the skin was cut in a circular manner. Once the skull was exposed, a local anesthetic (0.01–0.02 mL) made of NaCl supplemented with 1 mg/mL Lidocaine (Fresenius Kabi) and 0.25 mg/mL Bupivacaine (Actavis, Aurobindo Pharma B.V.) was applied. After removal of the periosteum, a 5 mm diameter circle was drilled on the right parietal bone. The bone was removed following removal of dura mater with thin forceps. The tumor cell suspension was injected in the middle of the craniotomy at 0.5 mm depth with a 10 µL glass Hamilton syringe, 30G and point 4-style needle at a speed of 1 µL/minute. Silicone oil covered the exposed brain and a 6 mm coverslip was glued on top with histoacryl (Braun). Dental acrylic cement (Vertex) was applied on the skull surface to glue a 3D printed window which provides fixation when performing intravital microscopy. After surgery, mice were allowed to recover on a heating pad and were closely monitored the following days.

### 2.7. Intravital Imaging Microscopy

Mice were sedated with 1% (*v*/*v*) isoflurane and placed in a custom-designed imaging box while kept under continuous anesthesia with isoflurane. Both the imaging box and microscope were adjusted to 36.5 °C using a climate chamber. During long-term imaging, mice were administered with 100 µL/h of Nutriflex special 70/240 (Braun). Intravital images were acquired with both an inverted Leica SP8 Dive system (Leica, Mannheim, Germany) with a MaiTai eHP DeepSee laser (Spectra-Physics, Milpitas, CA, USA) and an InSight X3 laser (Spectra–Physics) or an inverted Leica SP8 Dive system (Leica, Mannheim, Germany) with only InSight X3 laser (Spectra–Physics). For both systems a Leica Application Suite X (LAS X) software (Mannheim, Germany) was used. For the system with 2 lasers, SHG (crosslinked collagen I), E-cad-mCFP and H2B-Dendra2 were simultaneous excited with 860 nm (Mai Tai) and detected at 420–442 nm (HyD-RLD1), 444–485 nm (HyD-RLD2) and 510–556 nm (HyD-RLD3), respectively. ROSA26^mTmG^ was excited with 980 nm (Insight X3) and detected at 564–698 nm (HyD-RLD4). For the system with one laser, we performed sequential scanning; sequence 1 used an excitation of 930 nm (Insight X3) to detect SHG at 458–471 nm (HyD-RLD2), H2B-Dendra2 at 491–541 nm (HyD-RLD3) and ROSA26^mTmG^ at 564–684 nm (HyD-RLD4); sequence 2 used an excitation of 840 nm to image E-cad-mCFP at 442–475 nm (HyD-RLD3). Three-dimensional tile scans were acquired with a 5 μm z-step size. Time-lapse movies were acquired with a 5 μm z-step and imaged every hour for a total of 8 to 12 h. All images were acquired in 12 bit and with a 25×/0.95 NA water immersion objective. Dextran Texas Red 70,000 MW (Invitrogen, Cat. No. D1830) at a concentration of 10 mg/mL was intravenously injected prior imaging. Texas Red was excited at 980 nm and detected at 674–711 nm.

### 2.8. Movies Correction and xy Migration Analysis

Time-lapse movies were corrected for xyz drift movements using Huygens Professional software program (version 20.10, Hilversum, Netherlands). For this correction we used the ROSA26^mTmG^ channel as a reference. Per corrected position, we made a maximum projection of 3 consecutive z-planes throughout the complete z-stack for the analysis, thus covering the complete imaging depth. Cells were manually tracked using MTrackJ plugin from ImageJ. A total of 120 to 370 cells were tracked per mouse in 3 mice. For each population (cells in tumor lobes or in the brain parenchyma), 350 cells were analyzed. The XY position was determined over time by the MTrackJ plugin and the speed and persistence for each individual cell was calculated with Excel (Microsoft, Redmond, DC, USA). For each population (cells in tumor lobes or in the brain parenchyma), we determined the percentage of migratory cells based on the number of tracked cells which speed was equal or higher than 2.5 µm over the total number of tracked cells per population.

### 2.9. Western Blot

Protein lysates were made using 20 mmol/L Tris-HCl pH 8.0, 150 mmol/L NaCl, 1% NP-40, 10% Glycerol in milli-Q water, complemented with protease and phosphatase inhibitors (Roche, Cat. No. 11697498001). Protein lysates were quantified using the Pierce BCA protein assay kit (Thermo Scientific, Cat. No. 23225) and loaded on a NUPAGE 4–12% gel Bis-Tris gradient gel (Invitrogen, Cat. No. NP0321PK2). Transfer onto a nitrocellulose membrane (Bio-Rad, Cat. No. 1620112) was performed overnight in 1× transfer buffer (25 mmol/L Tris, 2 mol/L Glycine, 20% methanol in demineralized water). Membranes were blocked at room temperature for 1.5 h in 5% non-fat dry milk and incubated overnight at 4 °C with the antibodies diluted in 5% non-fat dry milk. Primary antibodies were ARPC3 (also denominated as p21-ARC; 1:1000; Invitrogen Cat. No. PA5-82253) and Tubulin (1:10 000; Abcam Cat. No. ab6160). Membranes were washed 3× with TBS-T and incubated for 1 h at room temperature with adequate secondary antibodies: anti-rabbit HRP (1:3000; BioRad Cat. No. 1706515) or anti-rat HRP (1:3000; Invitrogen Cat. No. 62-9520) in 5% non-fat dry milk. Membranes were washed 3× with TBS-T and developed using ECL-clarity (BioRad, Cat. No. 170-5060) on a Fusion FX7 edge (Vilber).

### 2.10. Intracranial Injection

BCBM organoids were prepared as mentioned in ‘Cranial window implantation and BCBM injection under the window’. Prior to surgery, mice received 0.067 mg/mL rimadyl (carprofen, Zoetis) in drinking water 1 day before injection and for 3 days post surgery. Additionally, 30 min before surgery and 8–12 h after surgery, mice were subcutaneously injected with 0.1 mg/kg temgesic (buprenorphine, Indivior Europe Limited). Mice were anesthetized via inhalation of 2% (*v*/*v*) isoflurane and eyes were covered with duratears (Alcon). After shaving the head, an incision was made in the skin to reveal the skull, followed by the application of a local anesthetic (0.01–0.02 mL). This was prepared with NaCl supplemented with 1 mg/mL Lidocaine (Fresenius Kabi) and 0.25 mg/mL Bupivacaine (Actavis, Aurobindo Pharma B.V.). After removal of the periosteum, a hole was drill 1.5 mm to the right and 1.5 mm posterior to the bregma. At a depth of 1 mm, we injected our tumor cells preparation (100,000 cells in 3 µL) with a 10 µL glass Hamilton syringe, 30G and point 4-style needle at a speed of 1 µL/minute. Skin was sutured after injection and the mouse was allowed to recover on a heating pad, being monitored closely the next days.

### 2.11. Magnetic Resonance Imaging

Magnetic resonance imaging was performed with a 7 Tesla BioSpec 70/20 USR (Bruker; Billerica, MA, USA). Imaging was performed with a T2 RARE sequence with a 39 ms echo-time, 2200 ms repetition time and 8 averages. Paravision software (v6.0.1; nBruker) was used for image acquisition and MIPAV NIH software (Bethesda, USA) was used for tumor volume measurements. Mice were anesthetized with 2% (*v*/*v*) isoflurane during imaging, eyes were covered with duratears (Alcon) and the heart rate was monitored throughout the procedure.

### 2.12. Tumor Resection

This surgery was designed in close collaboration with a neurosurgeon to resemble the clinical setting. During this entire experiment, the researcher performing the surgery and MRI was not aware of the BCBM group from the time of intracranial injection until the 11 weeks after tumor resection. Mice were intracranially injected with 20,000 cells derived from either shScramble or shARPC3 BCBM organoids. The coordinates from the bregma were 1.5 mm to the right, 1.5 mm posterior and 0.5 mm depth. Once mice developed tumors with a volume of 10–18 mm^3^, BCBM resection was performed. Mice received 0.067 mg/mL rimadyl (carprofen, Zoetis) in drinking water 1 day before surgical resection and for 3 days post surgery. Moreover, 30 min before surgery and 8 h after surgery, mice were subcutaneously injected with 0.1 mg/kg temgesic (buprenorphine, Indivior Europe Limited). Mice were anesthetized via inhalation of 2% (*v*/*v*) isoflurane and their eyes were covered with duratears (Alcon). The mouse head was shaved and Rymadil (5 mg/kg) and warm NaCl were subcutaneously injected. Once placed on the stereotactic frame, an incision was made in the skin to reveal the skull. A local anesthetic which consisted of NaCl supplemented with 1 mg/mL Lidocaine (Fresenius Kabi) and 0.25 mg/mL Bupivacaine (Actavis, Aurobindo Pharma B.V.) was applied on the skull. Removal of the periosteum was performed with a cotton swab, followed by drilling of a 5 mm diameter circle on the right parietal bone. Bone and meninges were removed with thin forceps. With a vacuum pump, the BCBM was aspirated with a 100 µL sterile tip until no more tumor was visible. To stop bleeding, we used cold NaCl with eyespear pointed tips (Sugi) and once mostly dried we applied spongstan (Ethicon). Once the bleeding stopped and the surgery finished, we used a fluorescent stereomicroscope system (Nightsea fluorescent adapter, Lexington, MA, USA) to visualize how much fluorescent tumor was left behind. Skin was sutured and glued with histoacryl (Braun). After surgery, mice received subcutaneously warm NaCl while recovering on a heating pad. Mice were closely monitored over the following days.

### 2.13. Fluorescence Activated Cell Sorting

BCBM were isolated from mouse brains and minced manually on ice using sterile scalpels. All of these steps were executed inside a flow cabinet. Minced tumors were collected in a 15 mL falcon and enzymatically digested in 3 mL PBS supplemented with 10 μg/mL DNase I (Roche, Cat. No. 10104159001) and 0.96 mg/mL Liberase TH (Roche, Cat. No. 5401151001) at 37 °C for 15 min at 1050 rpm. The remaining pieces were resuspended with a 20-gauge needle. Supernatant was spun down and washed twice with a FACS buffer made of PBS with 2 mmol/L EDTA (Lonza AccuGENE, Cat. No. 51234) and 1% B27 (50× Gibco, Cat. No. 17504044). After the last wash, the pellet was resuspended in FACS buffer and filtered through the lid of a falcon tube with a cell strainer (Falcon, Cat. No. 352235). Sample was transferred into a polypropylene falcon (Falcon, Cat. No. 352063) and just prior to sorting, 20 nmol/L of TO-PRO Iodide-3 642/661 was added (Invitrogen, Cat. No. T3605). Cells were sorted on a FacsAria Fusion 1 (BD Biosciences) (San Jose, USA) and the gating strategy is illustrated in Appendix A. The purity check control was performed using the CFP positive population since we did not have enough CFP negative cells for this purpose (Appendix A).

### 2.14. RNA Isolation, cDNA Synthesis and Real Time PCR (RT-qPCR)

RNA was isolated using TRIzol LS reagent (Invitrogen, Cat. No. 10296010) according to the manufacturer’s protocol. To help with the visualization of the pellet, GlycoBlue Coprecipitant (Invitrogen, Cat. No. AM9516) was used as described by the manufacturer’s protocol. The amount and purity of isolated RNA was analyzed with Nanodrop DS-11/DS-11+ Spectrophotometer (DeNovix, Wilmington, DE, USA). RNA was pre-treated with RNase free-Dnase (Promega, Cat. No. M6101) followed by cDNA synthesis with high-capacity cDNA Reverse Transcription kit (Applied Biosystems, Cat. No. 4368814, Waltham, MA, USA). Both reactions were performed according to the manufacturer’s instructions. qPCR was performed with PowerUp SYBR Green Master Mix (ThermoFischer Scientific, Cat. No. A25777) in a QuantStudio Real-Time qPCR machine (ThermoFischer Scientific, Waltham, MA, USA)and PPIA and RPL38 were used as housekeeping genes to enable normalization. Primers were used at a 150 nmol/L concentration and sequences are annotated in Appendix A. Thermal cycle conditions were as follows: 2 min at 50 °C, 2 min at 95 °C followed by 45 cycles consisting of denaturation for 15 s at 95 °C, annealing for 1 min at 60 °C, and extension for 1 min at 72 °C. PCR reactions were concluded with incubation for 10 min at 72 °C. RDML raw data was processed using RDML-Ninja (version 0_9_3 , Amsterdam, The Netherlands) and LinReg software (version 2020, Amsterdam, The Netherlands) to determine the baselines and obtain the empirical primer efficiencies. qPCR fold changes between samples were obtained by ΔΔCt calculations and corrected by primer efficiencies. Average values from the two housekeeping genes were calculated, normalized to E-cad^HI^ tumor cells, log2 transformed and the average plotted. Statistical significance was calculated with GraphPad Prism (San Diego, CA, USA) package using one sample *t*-test, one-tailed.

### 2.15. RNA Bulk Sequencing and Bioinformatics

Illumina TruSeq mRNA libraries were generated and sequenced with 65 base single reads on a HiSeq 2500 using V4 chemistry (Illumina Inc., San Diego, CA, USA) as previously described [37]. Reads were mapped to the genome with Hisat version 2.1.0. (Amsterdam, Netherlands) Read counts were determined with Icount based on the mouse reference genome GRCm38_87. To correct for sequencing depth variability between samples, a DESeq median-of-ratios approach was implemented [38] followed by a log-transformation of the normalized counts. These corrections were necessary in order to obtain the normalized expression values. The singscore package [39] was used to determine the EMT scores based on the Ecad^LO^ signature of PyMT breast tumors [27]. Statistical significance of both scores to 0 was calculated with one-sample *t*-test. DESeq2 [38] was used for the differential expression analysis. Pathway analysis was conducted with GSEA preranked [40] by using as metric -log10(FDR) divided by the sign of the log2FoldChange and the Hallmark gene sets as pathways [41].

Samples of the Beerling et al. [27] were downloaded from GEO number GSE77107. The reads were trimmed using Cutadapt [42] to remove any remaining adapter sequences and to filter reads shorter than 20 base pairs. The trimmed reads were aligned to the GRCm38 reference genome using STAR [43]. Quality control (QC) statistics from FastQC [44] and the above-mentioned tools were collected and summarized using MultiQC [45]. Gene expression counts were generated by featureCounts [46] using gene definitions from Ensembl GRCm38 version 100. Normalized expression values were obtained as described above. The entire analysis was implemented by Julian de Ruiter using Snakemake (snakemake version 5.19.2; wrapper version 0.63.0) (Amsterdam, Netherlands) [47] and is freely available on GitHub (https://github.com/jrderuiter/snakemake-rnaseq) (accessed on 4 June 2020).

### 2.16. Immunofluorescence and Confocal Imaging

Brain samples were fixed overnight in 4% periodate-lysine/4% paraformaldehyde [48] at 4 °C. The next day, tissues were washed with PBS, incubated overnight in 30% sucrose and frozen in OCT. Per tumor, 2 consecutive sections of 5 µm thickness were cut. After cutting, sections were hydrated twice with PBS for 15 min and mounted with anti-fading mounting medium (Vectashield, Vector laboratories (Cat. No. H-1000-10). Slides were imaged using inverted Leica TCS SP8 confocal microscope. H2B-Dendra2 was excited at 488 nm and collected at 495–545 nm. E-cad-mCFP signal was excited at 442 nm and collected at 454–479 nm with a line accumulation of 3. All images were acquired with a 25×/0.95 NA water immersion objective.

### 2.17. Immunohistochemistry

Brain samples were collected and fixed in formalin or EAF (ethanol/acetic acid/formaldehyde/saline at 40:5:10:45 *v*/*v*) and embedded in paraffin. Immunohistochemistry was performed on 4 µm-thick sections for E-cadherin antibody (Cell Signaling, Cat. No. 3195, 1:100) and KI67 antibody (Abcam, Cat. No. ab15580, 1:1000). Sections were imaged using an Aperio AT2 digital scanner with a 20× objective and processed with Aperio ImageScope software (both from Leica Biosystems, Wetzlar, Germany).

### 2.18. Human Samples and Multiplex Immunohistochemistry

All the human BCBM biospecimens were isolated at the Amsterdam UMC (location VUMC). Before the surgery of the brain tumors, patients at the A-UMC/VUMC are informed and have always had the opportunity to object to the use of their biospecimens. For all samples used in our study, patients did not object, and hence, they did therefore agree to the use of their biospecimens. This opt-out procedure is pursuant to the ethical rules and regulations of the Amsterdam UMC, location VUMC. Hence, the procedures comply both with (inter-) national legislative and ethical standards. In the database of the Amsterdam UMC, location VUMC, 8 samples of brain resections with a metastasis of mammary carcinoma were selected. These were all invasive ductal carcinomas, not otherwise specified, with a different receptor status: 2 ER-positive, 2 HER2-positive, 2 triple negative and 1 unknown receptor status. Immunohistochemistry of the Formalin Fixed Paraffin Embedded (FFPE) tumor samples was performed on a Discovery Ultra automated stainer (Ventana Medical Systems). Briefly, paraffin sections were cut with a 3 µm thickness, heated at 75 °C for 28 min and deparaffinized with an EZ prep solution (Ventana Medical Systems). Heat-induced antigen retrieval was carried out using Cell Conditioning 1 (CC1, Ventana Medical Systems) for 64 min at 95 °C. For the triple panel Zeb1 (DAB), Pankeratin (Yellow) and E-cadherin (Purple): Zeb1 was detected in the first sequence using clone E2G6Y (Cell Signaling, 1:200 dilution, 60 min at room temperature). Zeb1 bound antibody was visualized using Anti-Rabbit HQ (Ventana Medical systems) for 12 min at 37 °C followed by Anti-HQ HRP (Ventana Medical systems) for 12 min at 37 °C, followed by the ChromoMap DAB Detection kit (Ventana Medical Systems). In the second sequence of the triple staining procedure, Pankeratin was detected using clone AE1/AE3 (Abcam, 1:400 dilution, 60 min at 37 °C) Pankeratin was visualized using Anti-Mouse NP (Ventana Medical systems, Oro Valley, AZ, USA) for 12 min at 37 °C followed by Anti-NP AP (Ventana Medical systems) for 12 min at 37 °C, followed by Discovery Yellow Detection Kit (Ventana Medical Systems). For the third sequence, E-cadherin was detected using clone NCH-38 (Agilent/DAKO, 1:50 dilution, 60 min at 37 °C). E-cadherin was visualized using Anti-Mouse HQ (Ventana Medical systems) for 12 min at 37 °C followed by Anti-HQ HRP (Ventana Medical systems) for 12 min at 37 °C, followed by the Discovery Purple Detection Kit (Ventana Medical Systems). Slides were counterstained with Hematoxylin and Bluing Reagent (Ventana Medical Systems) and imaged with a 40× magnification with a PANNORAMIC^®^ 1000 scanner (3DHISTECH, Budapest, Hungary).

## 3. Results

### 3.1. Intravital Microscopy Identifies Migratory Single Cells at the Invasive Front of BCBM

To characterize BCBM cells that invade the brain parenchyma, we performed a total of 6 serial intracranial transplantations of tumor pieces originally derived from end-stage primary ductal carcinomas that had developed in the mammary glands of PyMT E-cad-mCFP reporter mice [27] (Figure 1A). Fast-growing brain metastases were established after the 6th round of transplantation in the brain. From these tumors, we derived organoids with the aim to maintain 3D heterogenic characteristics of brain metastases. From here on, tumors that develop upon intracranial injection of the single cell suspension of brain-enriched organoids are referred to as BCBMs. To study the behavior of BCBM cells in vivo, we intracranially injected these organoids labelled with a nuclear fluorescent marker (H2B-Dendra2) in ROSA26^mTmG^ FVB mice and monitored BCBM progression through a cranial imaging window using multi-photon microscopy. Two weeks after intracranial injection of BCBM organoids, we observed the formation of a tumor with individual lobes containing a high density of tumor cells (Figure 1B). Importantly, we observed single tumor cells outside of the tumor lobes (Figure 1B). These single tumor cells were associated with blood vessels (Figure 1B,C). To assess the behavior of these single cells, we performed time-lapse live imaging through the cranial imaging window for up to 12 h by multi-photon microscopy in ROSA26^mTmG^ mice (Figure 2A,B). We tracked the position of 350 cells within the tumor lobes and 350 cells at the invasive front over time in a total of 16 positions in 3 mice (Figure 2C,D). Whilst the vast majority of cells in the tumor lobes were non-migratory, a large proportion of the cancer cells in the brain parenchyma were motile (defined by cells with a speed equal to or higher than 2.5 µm per hour, Figure 2C,D). Moreover, cell migration in the parenchyma was highly persistent (Figure 2E). Additionally, we frequently observed cells migrating along structures such as type I collagen fibers as observed by second harmonic generation (SHG) (Figure 2F). Together, we identify a population of highly migratory cancer cells that escape from the BCBM lobes and invade the surrounding parenchyma.

### 3.2. Impeding Tumor Cell Migration Improves Surgical Resection

Considering that cancer cell invasiveness hurdles complete tumor resection and are associated with worse patient prognosis in BM and glioblastoma [49,50,51], we next tested whether impeding tumor cell migration would improve the efficiency of surgery in BCBM. Cell motility in BC, including PyMT breast tumors, was previously shown to be partly driven by the actin-related protein 2/3 complex (ARP2/3 complex), which mediates actin nucleation [52,53,54]. As such, reducing ARPC3 expression can impair cell migration without altering proliferation [55]. We therefore transduced our BCBM organoids with short-hairpin RNAs against ARPC3 (shARPC3). Western blot analysis confirmed the reduced expression of ARPC3 using two distinct shARPC3 (Figure 3A and Appendix A). For further experiments, we worked with shARPC3 #2, which showed the highest knockdown efficiency. Next, we intracranially injected BCBM organoids engineered with either shScramble or shARPC3 in ROSA26^mTmG^ FVB mice. We confirmed that reducing ARPC3 expression did not alter tumor cell proliferation, based on Ki67 expression (Figure 3B,C). Next, we used multi-photon microscopy to monitor BCBMs through a cranial window. In line with previous literature [52,53,55], reducing ARPC3 expression in BCBM cells significantly decreased the number of cells that invaded the surrounding parenchyma (Figure 3D,E).

Next, we tested whether impairing migration and invasion by ARPC3 knockdown would improve surgical removal of BCBM. We intracranially injected BCBM organoids engineered with either shScramble or shARPC3 in 16 FVB mice and monitored BCBM growth using MRI (Figure 4A). Whilst blinded for the experimental group, we performed surgical resection of the BCBM when tumors reached a volume of 10–18 mm3. The average tumor volume was equal between both groups (Figure 4B). Importantly, in line with the absence of significant changes in proliferation (Figure 3B,C), the time for tumors to reach the same average volume prior to resection did not differ between the shScramble and shARPC3 tumors, confirming that reducing ARPC3 expression does not influence tumor growth (Figure 4B,C). From the shARPC3 group, one mouse did not grow a BCBM and another mouse was sacrificed for BCBM-unrelated reasons; therefore, we excluded those mice from the experiment. Since our BCBM cells express H2B-Dendra2, we used fluorescent microscopy to score the amount of tumor cells that were left behind after surgical resection of the BCBMs (Figure 4D). This revealed that in all mice of the shScramble group, some BCBM cells remained in the surgical cavity; however, this was not the case in the majority of the shARPC3 mice (Figure 4D,E). Strikingly, even a week after surgical removal, those cells that were left behind were not detected by MRI (Figure 4F). This suggests that also in the clinic, where MRI is used as a measure of complete BCBM resection, single BCBM cells that invade the parenchyma and are left behind upon resection may not be detected. Additionally, similar to BCBM patients, we observed BCBM recurrence at later time-points with 50% of the shScramble mice recurring versus 33% in the shARPC3 group during the 11 weeks follow-up post-surgery (Figure 4F,G). Reduced expression of ARPC3 in the recurrent tumors was maintained, as confirmed by RT-qPCR (Appendix A). Together, these data suggest that blocking cell invasion may improve surgical resection of the BCBM.

### 3.3. BCBM Invading Tumor Cells Are in a Mesenchymal State

Next, we investigated the role of EMT in these invasive tumor cells using the E-cad-mCFP reporter, as we previously did in the primary tumor [26,27]. In the tumor lobes, cancer cells express E-cad-mCFP (Figure 5A). Importantly, we observed that all cells that invaded the parenchyma lost E-cad-mCFP expression (arrowheads, Figure 5A,B). From here on, we refer to cells with high E-cad-mCFP expression as E-cad^HI^ cells and cells that lost E-cad-mCFP expression as E-cad^LO^ cells. In line with our previous observations in primary PyMT breast tumors [26,27], E-cad^HI^ cells were on average non-motile, whilst a large fraction of the E-cad^LO^ cells were migratory (Figure 5C,D).

To characterize the Epithelial/Mesenchymal (E/M) states of E-cad^HI^ cells and E-cad^LO^ cells, we isolated both populations using fluorescence activated cell sorting (FACS) (Figure 5E and Appendix A). Imaging analysis of E-cad-mCFP confirmed the successful isolation of E-cad^HI^ and Ecad^LO^ cells (Appendix A). Next, we performed bulk mRNA sequencing on both cell populations (Figure 5F,G) and analyzed the differentially expressed genes that we refer to as the E-cad^LO^ gene set. The E-cad^LO^ gene set contains typical EMT genes, such as E-cadherin (Cdh1), N-cadherin (Cdh2), Vimentin (Vim), Fibronectin (Fn1) and transcription factors such as Snail (Snai1), Zeb1 and Zeb2 (Figure 5G). We confirmed the bulk mRNA sequencing results by RT-qPCR (Figure 5G). In line with this data, the E-cad^LO^ gene set was enriched for genes involved in the “Hallmark Epithelial_Mesenchymal_Transition” [42] (Figure 5H). Lastly, we compared the expression profile of E-cad^LO^ cells isolated from BCBM to the expression profile of their counterparts isolated from primary PyMT tumors. Importantly, in our prior study, we showed that E-cad^LO^ cells have low expression levels of the epithelial marker E-cadherin and high expression of the mesenchymal marker Vimentin [27]. In line with the enrichment of genes involved in EMT, we found that the E-cad^LO^ gene set but not E-cad^HI^ gene set of BCBM tumors is enriched in genes that were found to be significantly upregulated in the E-cad^LO^ gene set of the primary breast tumor [27]. Collectively, these data show that the differential expression of E-cad-mCFP in E-cad^HI^ and E-cad^LO^ cells reports the epithelial and mesenchymal states of cells, respectively, in our PyMT BCBM model.

### 3.4. Mesenchymal Cells Can Form New Tumors and Revert to an Epithelial State

Since the presence of invasive BCBM cancer cells is involved in relapse following surgical resection (Figure 4 and [49,51]), we next investigated whether the E-cad^LO^ cells have the capacity to grow BCBMs. We isolated~2000 E-cad^LO^ cells from BCBM grown in ROSA26^mTmG^ FVB mice by FACS and injected this population intracranially in tumor free FVB mice (Figure 6A). An assessment of the purity suggested a potential contamination of the sorted fraction of at most 2.5% with other cells. Therefore, we injected 2.5% of E-cad^HI^ cells in tumor free mice to serve as a purity control group (Appendix A). In 75% of mice injected with E-cad^LO^ cells, BCBMs had developed during approximately 7 weeks post-intracranial injection, while no tumors developed during the same follow-up period in the purity control group (Figure 6B). Next, we isolated the BCBMs generated by the E-cad^LO^ cells and studied the expression of the E-cad-mCFP reporter (Figure 6C). In all tumors, we detected the E-cad-mCFP reporter in tumor cells, which was further validated by staining tumors with an E-cadherin antibody (Figure 6C). This demonstrated that in our BCBM model, mesenchymal cells have the ability to form new tumors and to concomitantly revert to an epithelial phenotype, a process often referred to as mesenchymal-to-epithelial transition (MET).

### 3.5. Tumor Cells Undergoing EMT Are Present at the Invasion Front in Human BCBM

Finally, we investigated whether mesenchymal infiltrative tumor cells are also present in BCBM patients (Figure 7). We obtained tumor resections from 8 patients with BCBM. Using multiplex immunohistochemistry, we simultaneously stained for Pankeratin and E-cad to mark cells in an epithelial state, and Zeb1 to mark cells in a mesenchymal state (Figure 7A). In the tumor mass, we indeed observed that cancer cells stain positive for Pankeratin and E-cad and negative for Zeb1, as expected for epithelial cancer cells. Moreover, and in line with previous observations [56], the brain parenchyma that surrounds the BCBM contained cells that were positive for Zeb1 and negative for E-cad. Yet, most of these cells were also negative for Pankeratin, suggesting that those cells belong to a population of glial and immune cells that also express Zeb1 [57]. However, in seven out of eight patients, we also observed a population of Zeb1 positive; E-cad negative cells with moderate to high expression of the epithelial marker Pankeratin (Figure 7B). In line with the data of the infiltrative E-cad^LO^ cells that are present in the brain parenchyma in our mouse model, these Pankeratin-positive, Zeb1-positive, E-cad-negative cells have a spindle shape nucleus, indicative of a migratory state (encircled in Figure 7A). Together, this suggests that BCBM patients also have infiltrative tumor cells in EMT in the brain parenchyma that surrounds the BCBM.

## 4. Conclusions

The role of EMT in the formation of metastases has been heavily debated by the field. However, the role of EMT at later stages of the metastatic cascade in growing metastatic masses remains under-studied. Here, we used intravital microscopy combined with mRNA sequencing to demonstrate that cells undergoing EMT are present in established BCBM, invade the brain parenchyma and have the capacity to revert to an epithelial state when forming secondary lesions. We also reveal that the presence of these invasive cells impairs the successful removal of BCBM lesions. Importantly, even if surgery is followed by chemotherapy or radiotherapy, these invasive cells may not be efficiently depleted, since EMT endow cells with properties that can render them resistant to those therapies [21,58,59,60]. Therefore, targeting EMT can improve surgical resection and may sensitize cells that are left behind to subsequent therapies.

In our study, we combined an E-cad fluorescent reporter with mRNA sequencing to study EMT in BCBM. We showed that the E-cad^LO^ cells upregulate mesenchymal markers and are migratory. However, recent work has provided evidence that EMT is not a binary switch but that cells can adopt various hybrid E/M states with distinct metastatic and chemoresistance properties [17,19,25,61,62]. Here, in our mouse experiments, we have neither investigated the presence nor the role of cells with different E/M states. However, in our analysis of human BCBM, we observed cancer cells that escaped from the tumor mass and invaded the brain parenchyma contained a typical mesenchymal profile of expressing Zeb1 and losing E-cadherin, but interestingly still expressing the epithelial marker Pankeratin. Expression of the latter marker suggests that these infiltrating cells may not have lost all epithelial traits which can indicate a hybrid state. Future studies should shed light on the role of specific hybrid E/M states in BCBM and their distinct contributions to patient relapse following surgery. Additionally, developing therapeutic strategies to block EMT and/or mesenchymal cells is required to target this process and to ultimately improve BCBM patients’ outcomes. Nevertheless, our data warrant future studies to target EMT processes for better treatment of BCBM.

## Figures and Tables

**Figure 1 cancers-14-03115-f001:**
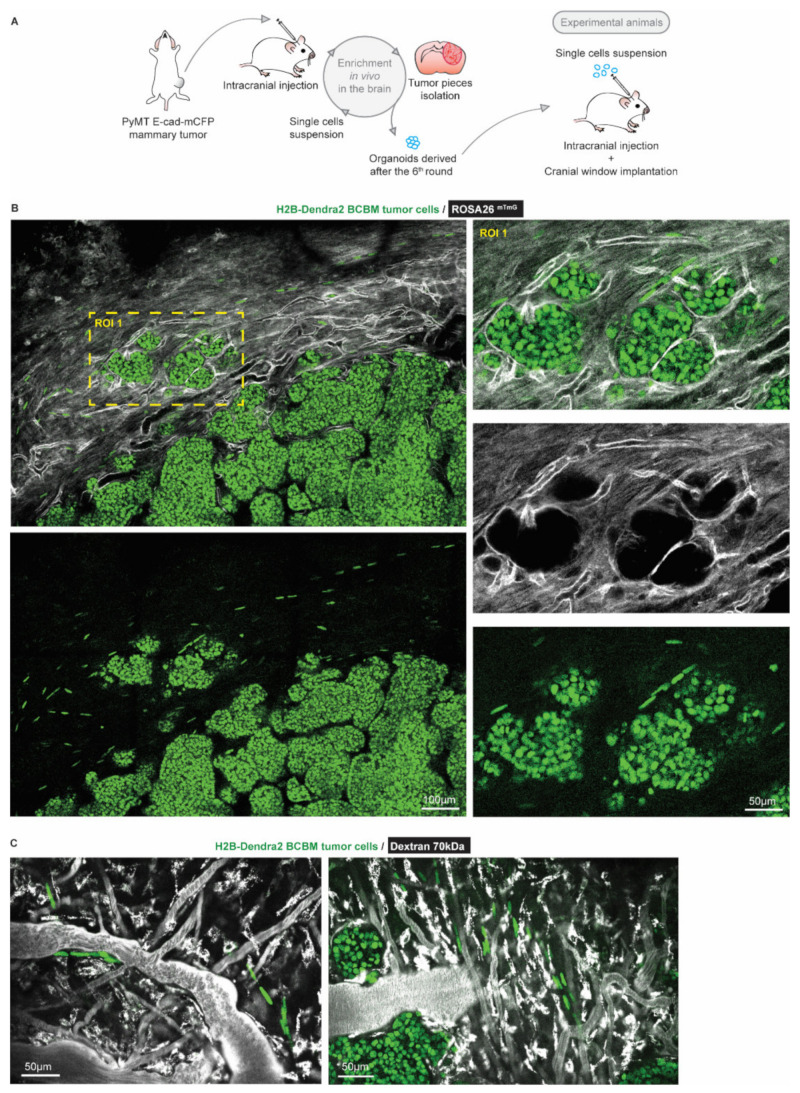
Intravital microscopy reveals the presence of single tumor cells at the invasive front of BCBM. (**A**) Schematic representation of the generation of a BCBM model derived from PyMT, E-cad-mCFP tumor. (**B**) Representative image (*n* = 4 mice) obtained by multi-photon imaging through a cranial window of a BCBM invasive front, 2 weeks after tumor implantation. The image depicts tumor cells expressing H2B-Dendra2 (green) grown in ROSA26^mTmG^ mice. The mouse brain parenchyma expressed tdTomato (mTmG; grey). The invasion front shows the presence of single tumor cells expressing H2B-Dendra2 (green) that are disconnected from the tumor lobules and associated with the brain vasculature (mTmG, grey). (**C**) Representative pictures obtained by multi-photon imaging through a cranial window of BCBM 2 weeks after tumor implantation. Images show single tumor cells expressing H2B-Dendra2 (green) in close association with blood vessels. Blood vessels were imaged with 70 kDa fluorescent-dextran (grey).

**Figure 2 cancers-14-03115-f002:**
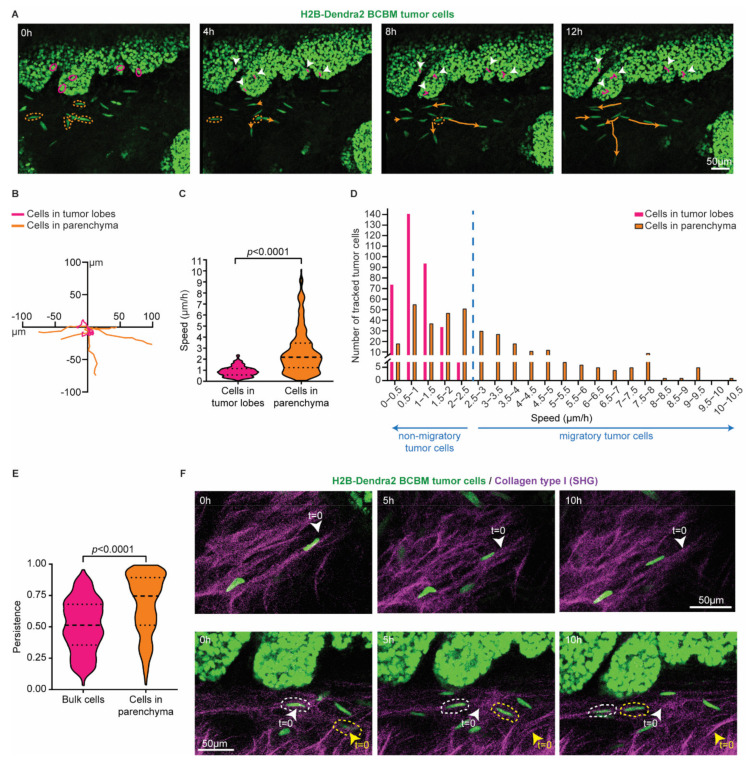
Single cells at the invasive front of BCBM are highly migratory. (**A**) Representative still images from a time-lapse multi-photon tracking of BCBM through a cranial window. BCBM tumor cells expressing H2B-Dendra2 were imaged every hour for up to 12 h. Randomly picked cancer cells located in the tumor lobe or brain parenchyma are marked with a circle in magenta and orange, respectively. Tracks are depicted once cells move. Migratory tracks depict the total movement of tumor cells in the tumor lobes (magenta tracks and white arrowheads) and brain parenchyma (orange tracks). (**B**) Rose plot representation of the migratory tracks with a common origin of tumor cells displayed in (**A**). (**C**) Quantification of the speed of tumor cells in the tumor lobes (magenta) or brain parenchyma (orange); *n* = 350 tumor cells per condition imaged in 3 mice; *p*-value was determined using a Mann–Whitney *t*-test, two-tailed. (**D**) Distribution of tumor cell speed for cells present in the tumor lobes and in the brain parenchyma. (**E**) Quantification of the persistence of migratory tracks from a total of 350 tumor cells per condition from 3 mice. *p*-value was determined with a Mann–Whitney *t*-test, two-tailed. (**F**) Representative still images of a time-lapse multi-photon movie showing the migration of BCBM tumor cells along crosslinked collagen type I fibers (SHG channel) in two distinct areas.

**Figure 3 cancers-14-03115-f003:**
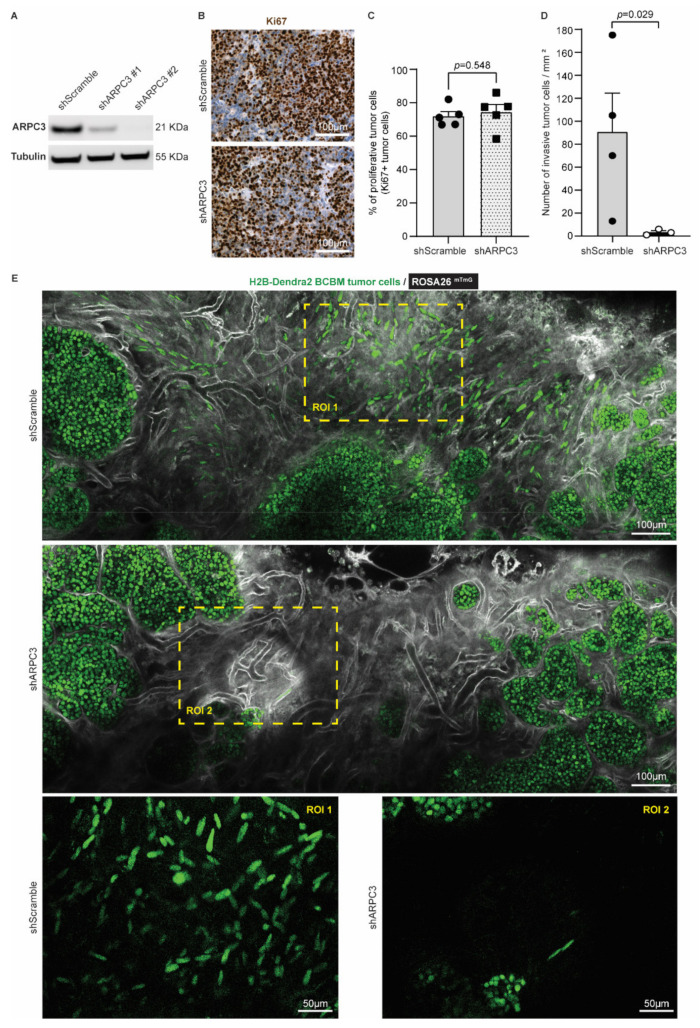
Reducing ARPC3 expression decreases the number of invasive tumor cells. (**A**) Western-blot of ARPC3 and Tubulin on BCBM tumor cells engineered with the shScramble or shARPC3 constructs. See Appendix A for uncropped blot. (**B**) Representative images and (**C**) quantification of Ki67 immunohistochemistry staining in BCBM derived from shScramble or shARPC3 organoids intracranially injected into recipient mice. Data are presented as mean +/− standard-error of the mean (SEM); *n* = 5 mice per group; *p*-value was determined using a Mann–Whitney *t*-test, two-tailed. (**D**) Quantification of the number of invasive tumor cells observed by intravital imaging of ROSA26^mTmG^ mice injected with shScramble or shARPC3 BCBM organoids. Data are presented as mean +/− SEM; *n* = 4 mice for shScramble and *n* = 3 mice for shARPC3 #2; *p*-value was determined using a Mann–Whitney *t*-test, one-tailed. (**E**) Representative multi-photon images of BCBM derived from shScramble and shARPC3 BCBM organoids intracranially injected into ROSA26^mTmG^ mice.

**Figure 4 cancers-14-03115-f004:**
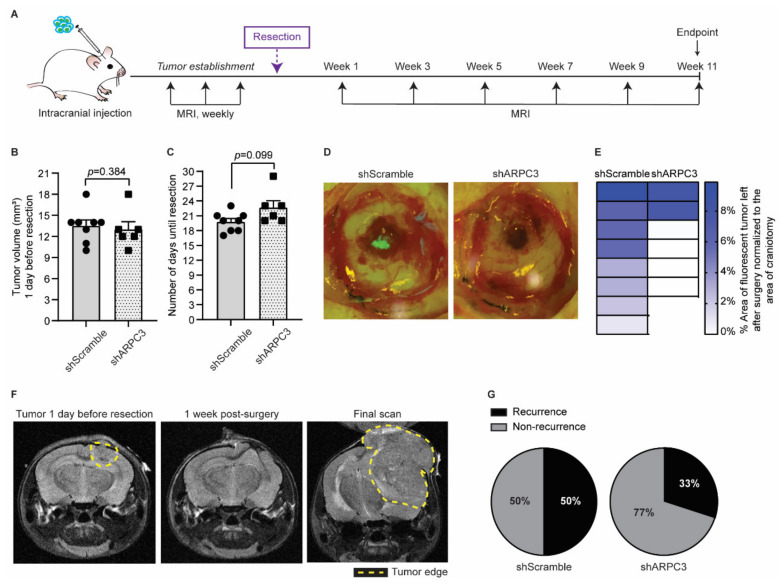
Impeding tumor cells migration improves surgical resection. (**A**) Schematic representation of experimental design. (**B**) Quantification of the tumor volume in mice intracranially injected with BCBM organoids engineered with the shScramble (*n* = 8) and shARPC3 (*n* = 6) constructs, one day before resection and as measured by MRI. Data are presented as mean +/− SEM; *p*-value was determined using a Mann–Whitney *t*-test, two-tailed. (**C**) Time between intracranial injection and resection for mice injected with BCBM organoids engineered with the shScramble (*n* = 8) and shARPC3 (*n* = 6) constructs. Data are presented as mean +/− SEM; *p*-value was determined using a Mann–Whitney *t*-test, two-tailed. (**D**) Representative images and (**E**) quantification of the percentage of tumor area (green fluorescent signal of H2B-Dendra2 expressed by all BCBM cells) left after resection. Each square represents one mouse. (**F**) Representative MRI images of a mouse brain at the various stages of the experiment. (**G**) Pie charts showing the percentage of mice that recurred until 11 weeks after resection. Additionally, see Appendix A.

**Figure 5 cancers-14-03115-f005:**
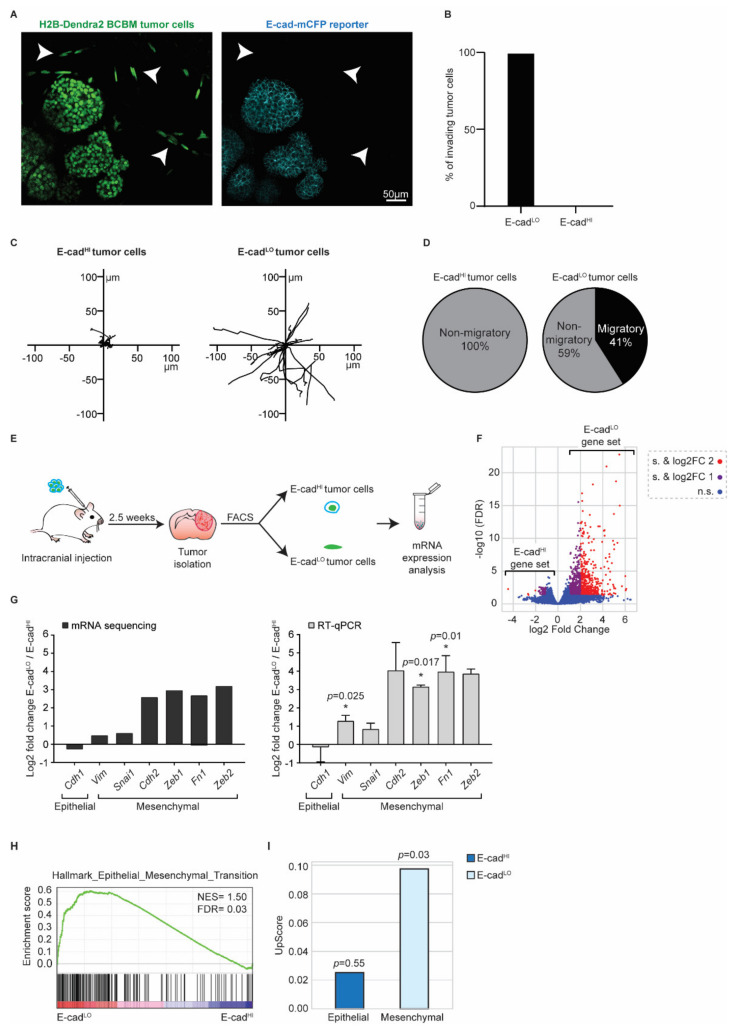
BCBM invading tumor cells are in a mesenchymal state. (**A**) Representative multi-photon images of BCBM tumor cells expressing H2B-Dendra2 (green) and endogenous E-cad-mCFP (turquoise). White arrowheads point to invading tumor cells. (**B**) Percentage of invading tumor cells that are E-cad^LO^ and E-cad^HI^. (**C**) Representative rose plots of migratory tracks with a common origin of 20 E-cad^HI^ tumor cells and 20 E-cad^LO^ tumor cells imaged using multi-photon imaging through a cranial window. (**D**) Pie charts of migratory (speed ≥ 2.5 µm/h) E-cad^HI^ or E-cad^LO^ tumor cells. (**E**) Schematic representation of experimental design used for isolating E-cad^HI^ or E-cad^LO^ tumor cells for mRNA sequencing and RT-qPCR. (**F**) Volcano plot comparing the fold change and false discovery rate (FDR) of E-cad^HI^ and E-cad^LO^ BCBM tumor cells. (**G**) mRNA expression analysis of EMT-related genes by mRNA sequencing and RT-qPCR; *n* = 4 for mRNA sequencing and *n* = 3 for RT-qPCR with exception for Cdh2 and Zeb2, which are *n* = 2. Data are presented as mean +/− SEM and standard deviation (SD) for Cdh2 and Zeb2. *p*-values were determined with a one-sample *t*-test, one-tailed between E-cad^LO^ and E-cad^HI^ tumor cells. (**H**) Gene set enrichment analysis [41] showing enrichment of the signature “Hallmark Epithelial_Mesenchymal_Transition” in E-cad^LO^ BCBM tumor cells. (**I**) Upscore enrichment of E-cad^HI^ or E-cad^LO^ BCBM tumor cells in E-cad^LO^ breast tumor cells [27]. Value of 0 implies that there is no enrichment of one gene set over the other. *p*-values were determined with a one-sample *t*-test to calculate the significance to 0.

**Figure 6 cancers-14-03115-f006:**
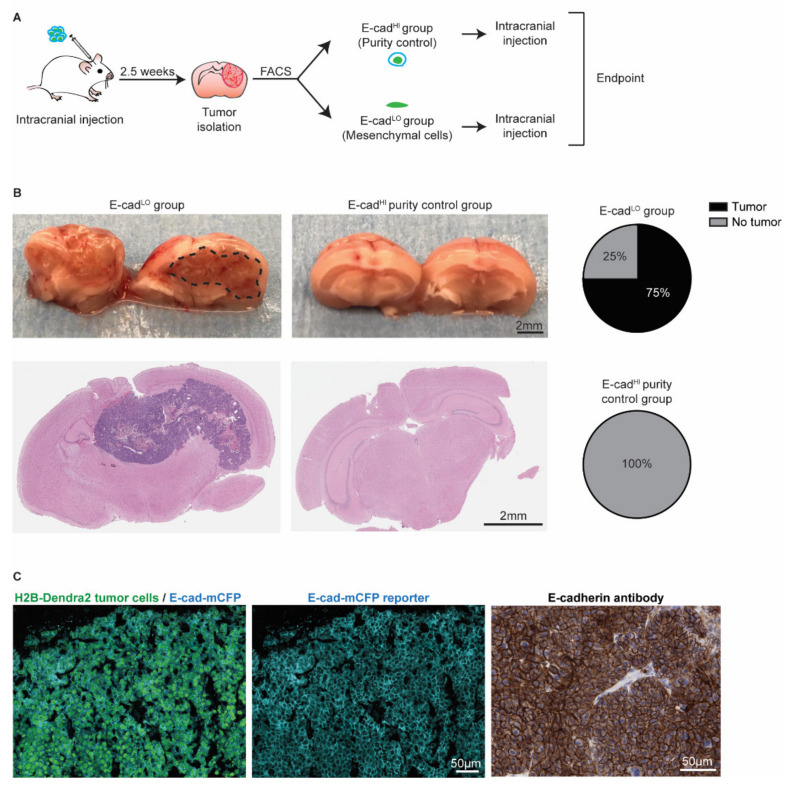
Mesenchymal cells can grow tumors and revert to an epithelial state. (**A**) Schematic representation of experimental design. (**B**) Representative pictures and H&E staining of the mouse brain. Pie charts show the percentage of BCBM tumors formed after intracranial injection of approximately 2000 E-cad^LO^ tumor cells compared to the purity control group. *n* = 4 mice per group. (**C**) Representative images of immunofluorescent staining (left and middle panels) of H2B-Dendra2 (green) and the endogenous E-cad-mCFP (turquoise) reporter in tumors formed after intracranial injection of E-cad^LO^ tumor cells. The expression of E-cadherin was confirmed using immunohistochemistry with an E-cadherin antibody (right panel). *n* = 3 mice.

**Figure 7 cancers-14-03115-f007:**
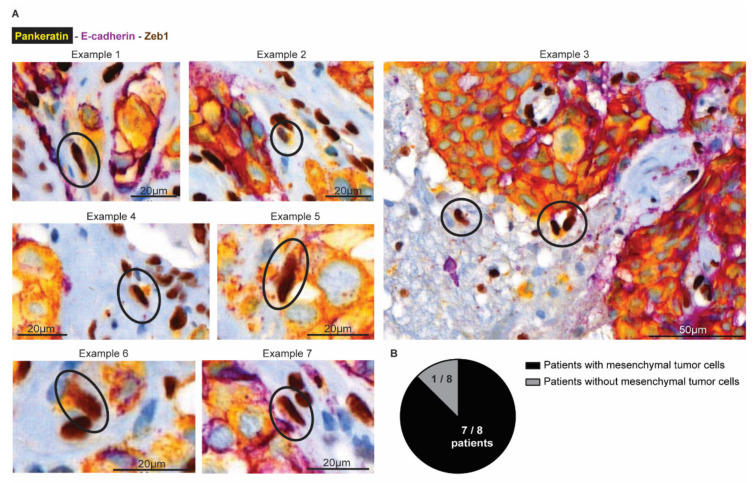
Mesenchymal tumor cells are present at the invasive front in human BCBM. (**A**) Multiplex immunohistochemistry imaging of Pankeratin (yellow), E-cadherin (purple) as markers for epithelial cells and Zeb1 (DAB) to mark cells in a mesenchymal state. (**B**) Quantification of the number of patients where mesenchymal tumor cells were observed in the brain parenchyma.

## Data Availability

The mRNA sequencing data are available through GEO number GSE199775. GEO Accession viewer (nih.gov).

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
