# Peer review of "Epithelial-to-Mesenchymal Transition Drives Invasiveness of Breast Cancer Brain Metastases"

_cancers, 2022, doi:10.3390/cancers14133115_

Round 1

Reviewer 1 Report

The article proposed by the authors is generally an original and complex scientific study. The selected methods are reliable and relevant to the research problem.

The most significant and original results of the study include the following: Intravital Microscopy Identifies Migratory Single Cells at the Invasive Front of BCBM; Impeding Tumor Cell Migration Improves Surgical Resection; BCBM Invading Tumor Cells are in a Mesenchymal State; Mesenchymal Cells can form New Tumors and Revert to an Epithelial State; Tumor Cells Undergoing EMT are Present at the Invasion Front in Human BCBM.

I believe that the article deserves to be published.

Author Response

Reviewer 1:

The article proposed by the authors is generally an original and complex scientific study. The selected methods are reliable and relevant to the research problem.

The most significant and original results of the study include the following: Intravital Microscopy Identifies Migratory Single Cells at the Invasive Front of BCBM; Impeding Tumor Cell Migration Improves Surgical Resection; BCBM Invading Tumor Cells are in a Mesenchymal State; Mesenchymal Cells can form New Tumors and Revert to an Epithelial State; Tumor Cells Undergoing EMT are Present at the Invasion Front in Human BCBM.

I believe that the article deserves to be published.

Reply:

We thank this referee for his/her time to review our manuscript. We are thankful for the support of our manuscript and highly appreciate his/her comments that our study is reliable and relevant to the research problem.

Reviewer 2 Report

The authors described EMT transition in the breast cancer brain metastasis by using intravital microscopy combined with mRNA sequencing. Overall, this is an interesting study demonstrating some promising results. Here are some suggestions.

1.  The authors claimed that reducing ARPC3 expression decreases the number of invasive tumor cells. (Fig 3) 

Would the ARPC3 knockdown cells show different phenotypes of the migration/invasion property? (Wound healing or transwell)

2. The EMT-related protein level should be investigated between the E-cad-low and E-cad-high groups. (e.g. Vimentin, Slug, Snail etc.)

3. In Figure 5G, the authors confirmed their RNA-seq data with the qPCR assay. Why do you need the t-test to compare the difference between RNA-seq and qPCR. The correlation analysis would be more appropriate if you want to demonstrate they are similar.

Reviewer 3 Report

The introduction provides a very informative and well-researched overview of brain metastases and the putative (and likely) tumor cell plasticity, in particular, EMT, may play in it. The references cited in this part appear fully comprehensive and sufficient to provide the reader with the necessary background. 

The experimental part, in my opinion, provides a brilliant example of tumor cell plasticity, and adaptation to the tumor microenvironment. 

The successive generation of brain "metastases", and selection of cells and later organoids with increasing capacity for generating metastases by intracranial injection of isolated tumor cell suspensions is of course somewhat artificial but justified. - there are no spontaneous brain metastasis models. Based on which aspects was decided that organoids are used for metastasis formation, and not single-cell suspensions ? (Why after the 6th round, not the 5th or the 7th?). That could be explained to the interested reader. 

I personally find the spectrum of experimental methods satisfying and comprehensive, especially the parallel production of organoids, and genetic modification of organoid (cells) by lentivirus gene transfer. This is not done routinely in many laboratories as of yet, but maybe it should. 

Last not least, cranial injection of tumor cell suspensions into the brain - followed by intravital imaging - clearly are two exceedingly complex techniques that both require excellent mastership, and seem to be expertly done here. I wonder if all mice survived the surgical procedures, though... this accounts both for the implantation of tumors cells/organoids, and the surgical removal of metastatic lesions later on. 

The description of bioinformatic procedures is a bit "slangy", and probably difficult to understand for readers that are not at home in the field of transcriptomics data analyses; this could be formulated a bit less cryptic. Otherwise, the materials & methods part is very detailed (altogether, 7 pages!) and appears to cover even smaller issues of the procedures used. 

Results: I cannot find any flaws with the intravital imaging and the presentation of representative images in Fig. 1 and 2; to illustrate various aspects of tumor cell invasion and motility of cells at the invasive front, contrasted by the sessile nature of the tumor cells within lobes. The quality and selection of these images is very convincing and (as with everything else here) expertly done. These figures are also not exceedingly complex, despite the complex nature of the observations. 

I also like the connection of experimental data (e.g., with surgical removal of tumors formed by cells with impaired tumor cell motility) to clinical questions. However, I also wonder, wouldn't organoids with impaired motility (due to the loss of ARPC3 expression) also form metastases less effectively in the mouse brains (see Fig. 3B versus 3C and 3D)? How do the authors explain that this is not the case? How does ARPC3 so effectively and exclusively affect tumor cell motility without having any effect on metastasis formation and cell proliferation? 

The data from Fig. 5 are almost of the kind of "see, I told you so..." : EMT is a phenotypic driver of tumor cell motility and invasion. As such, it is of course at the same time a brilliant demonstration of the role of EMT-genes in tumor cell plasticity. It also contains, on the other hand, few surprises as these things have been postulated and demonstrated before; also elsewhere. But that should of course not be held against the authors. I just wonder if this observation, and the E-cad HI/LO system could also be used for NEW discoveries. 

The icing on the cake are he validating experiments with a small selection of human brain metastasis lesions, essentially confirming some of the observations from the mouse experiments. 

In the short discussion, the authors speak (very briefly) of "targeting EMT for better treatment of brain metastasis".... what would be a reasonable, or actionable, target in their opinion - in connection with the data presented here? 

Reviewer 4 Report

In their study, the authors addressed the problem of breast cancer metastases to the brain. The authors designed the research in a thoughtful and thorough manner. In my opinion, the performance of the experiments and the description of the results do not raise any objections. The obtained results and conclusions drawn by the authors may be a significant contribution to the development of research on BC metastases to the brain in many patients. In conclusion, I recommend the submitted manuscript for publication in Cancers.

Author Response

Reviewer 4:

In their study, the authors addressed the problem of breast cancer metastases to the brain. The authors designed the research in a thoughtful and thorough manner. In my opinion, the performance of the experiments and the description of the results do not raise any objections. The obtained results and conclusions drawn by the authors may be a significant contribution to the development of research on BC metastases to the brain in many patients. In conclusion, I recommend the submitted manuscript for publication in Cancers.

Reply:

We thank this referee for his/her time to review our manuscript and for the recommendation to publish our manuscript with Cancers.

Round 2

Reviewer 2 Report

The authors have addressed the concerns and the manuscript can be considered for publication.